# Semi-supervised Sequence Learning

**Andrew M. Dai**
Google Inc.
adai@google.com

**Quoc V. Le**
Google Inc.
qvl@google.com

## Abstract

We present two approaches to use unlabeled data to improve Sequence Learning
with recurrent networks. The first approach is to predict what comes next in a
sequence, which is a language model in NLP. The second approach is to use a
sequence autoencoder, which reads the input sequence into a vector and predicts
the input sequence again. These two algorithms can be used as a "pretraining"
algorithm for a later supervised sequence learning algorithm. In other words, the
parameters obtained from the pretraining step can then be used as a starting point
for other supervised training models. In our experiments, we find that long short
term memory recurrent networks after pretrained with the two approaches be-
come more stable to train and generalize better. With pretraining, we were able to
achieve strong performance in many classification tasks, such as text classification
with IMDB, DBpedia or image recognition in CIFAR-10.

## 1 Introduction

Recurrent neural networks (RNNs) are powerful tools for modeling sequential data, yet training
them by back-propagation through time [37, 27] can be difficult [9]. For that reason, RNNs have
rarely been used for natural language processing tasks such as text classification despite their ability
to preserve word ordering.

On a variety of document classification tasks, we find that it is possible to train an LSTM [10] RNN
to achieve good performance with careful tuning of hyperparameters. We also find that a simple
pretraining step can significantly stabilize the training of LSTMs. A simple pretraining method is
to use a *recurrent language model* as a starting point of the supervised network. A slightly better
method is to use a *sequence autoencoder*, which uses a RNN to read a long input sequence into
a single vector. This vector will then be used to reconstruct the original sequence. The weights
obtained from pretraining can then be used as an initialization for the standard LSTM RNNs. We
believe that this semi-supervised approach [1] is superior to other unsupervised sequence learning
methods, e.g., Paragraph Vectors [19], because it can allow for easy fine-tuning.

In our experiments with document classification tasks with 20 Newsgroups [17] and DBpedia [20],
and sentiment analysis with IMDB [22] and Rotten Tomatoes [26], LSTMs pretrained by recurrent
language models or sequence autoencoders are usually better than LSTMs initialized randomly.

Another important result from our experiments is that it is possible to use unlabeled data from re-
lated tasks to improve the generalization of a subsequent supervised model. For example, using
unlabeled data from Amazon reviews to pretrain the sequence autoencoders can improve classifi-
cation accuracy on Rotten Tomatoes from 79.0% to 83.3%, an equivalence of adding substantially
more labeled data. This evidence supports the thesis that it is possible to use unsupervised learning
with more unlabeled data to improve supervised learning. With sequence autoencoders, and outside
unlabeled data, LSTMs are able to match or surpass previously reported results.

Our semi-supervised learning approach is related to Skip-Thought vectors [14], with two differences. The first difference is that Skip-Thought is a harder objective, because it predicts adjacent sentences. The second is that Skip-Thought is a pure unsupervised learning algorithm, without fine-tuning.

## 2   Sequence autoencoders and recurrent language models

Our approach to sequence autoencoding is inspired by the work in sequence to sequence learning (also known as *seq2seq*) by Sutskever et al. [32], which has been successfully used for machine translation [21, 11], text parsing [33], image captioning [35], video analysis [31], speech recognition [4] and conversational modeling [28, 34]. Key to their approach is the use of a recurrent network as an encoder to read in an input sequence into a hidden state, which is the input to a decoder recurrent network that predicts the output sequence.

The sequence autoencoder is similar to the above concept, except that it is an unsupervised learning model. The objective is to reconstruct the input sequence itself. That means we replace the output sequence in the *seq2seq* framework with the input sequence. In our sequence autoencoders, the weights for the decoder network and the encoder network are the same (see Figure 1).

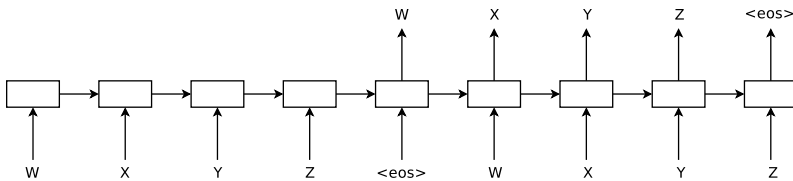

Figure 1: The sequence autoencoder for the sequence "WXYZ". The sequence autoencoder uses a recurrent network to read the input sequence in to the hidden state, which can then be used to reconstruct the original sequence.

We find that the weights obtained from the sequence autoencoder can be used as an initialization of another supervised network, one which tries to classify the sequence. We hypothesize that this is because the network can already memorize the input sequence. This reason, and the fact that the gradients have shortcuts, are our hypothesis of why the sequence autoencoder is a good and stable approach in initializing recurrent networks.

A significant property of the sequence autoencoder is that it is unsupervised, and thus can be trained with large quantities of unlabeled data to improve its quality. Our result is that additional unlabeled data can improve the generalization ability of recurrent networks. This is especially useful for tasks that have limited labeled data.

We also find that recurrent language models [2, 24] can be used as a pretraining method for LSTMs. This is equivalent to removing the encoder part of the sequence autoencoder in Figure 1. Our experimental results show that this approach works better than LSTMs with random initialization.

## 3   Overview of baselines

In our experiments, we use LSTM recurrent networks [10] because they are generally better than RNNs. Our LSTM implementation is standard and has input gates, forget gates, and output gates [6, 7, 8]. We compare this basic LSTM against a LSTM initialized with the sequence autoencoder method. When the LSTM is initialized with a sequence autoencoder, the method is called SA-LSTM in our experiments. When LSTM is initialized with a language model, the method is called LM-LSTM. We also compare our method to other baselines, e.g., bag-of-words methods or paragraph vectors, previously reported on the same datasets.

In most of our experiments our output layer predicts the document label from the LSTM output at the last timestep. We also experiment with the approach of putting the label at every timestep and linearly increasing the weights of the prediction objectives from 0 to 1 [25]. This way we can inject gradients to earlier steps in the recurrent networks. We call this approach *linear label gain*.

Lastly, we also experiment with the method of jointly training the supervised learning task with the sequence autoencoder and call this method *joint training*.

## 4 Experiments

In our experiments with LSTMs, we follow the basic recipes as described in [7, 32] by clipping the cell outputs and gradients. The benchmarks of focus are text understanding tasks, with all datasets being publicly available. The tasks are sentiment analysis (IMDB and Rotten Tomatoes) and text classification (20 Newsgroups and DBpedia). Commonly used methods on these datasets, such as bag-of-words or n-grams, typically ignore long-range ordering information (e.g., modifiers and their objects may be separated by many unrelated words); so one would expect recurrent methods which preserve ordering information to perform well. Nevertheless, due to the difficulty in optimizing these networks, recurrent models are not the method of choice for document classification.

In our experiments with the sequence autoencoder, we train it to reproduce the full document after reading all the input words. In other words, we do not perform any truncation or windowing. We add an end of sentence marker to the end of each input sequence and train the network to start reproducing the sequence after that marker. To speed up performance and reduce GPU memory usage, we perform truncated backpropagation up to 400 timesteps from the end of the sequence. We preprocess the text so that punctuation is treated as separate tokens and we ignore any non-English characters and words in the DBpedia text. We also remove words that only appear once in each dataset and do not perform any term weighting or stemming.

After training the recurrent language model or the sequence autoencoder for roughly 500K steps with a batch size of 128, we use both the word embedding parameters and the LSTM weights to initialize the LSTM for the supervised task. We then train on that task while fine tuning both the embedding parameters and the weights and use early stopping when the validation error starts to increase. We choose the dropout parameters based on a validation set.

Using SA-LSTMs, we are able to match or surpass reported results for all datasets. It is important to emphasize that previous best results come from various different methods. So it is significant that one method achieves strong results for all datasets, presumably because such a method can be used as a general model for any similar task. A summary of results in the experiments are shown in Table 1. More details of the experiments are as follows.

Table 1: A summary of the error rates of SA-LSTMs and previous best reported results.

| Dataset | SA-LSTM | Previous best result |
|---|---|---|
| IMDB | 7.24% | 7.42% |
| Rotten Tomatoes | 16.7% | 18.5% |
| 20 Newsgroups | 15.6% | 17.1% |
| DBpedia | 1.19% | 1.74% |

### 4.1 Sentiment analysis experiments with IMDB

In this first set of experiments, we benchmark our methods on the IMDB movie sentiment dataset, proposed by Maas et al. [22].[1] There are 25,000 labeled and 50,000 unlabeled documents in the training set and 25,000 in the test set. We use 15% of the labeled training documents as a validation set. The average length of each document is 241 words and the maximum length of a document is 2,526 words. The previous baselines are bag-of-words, ConvNets [13] or Paragraph Vectors [19].

Since the documents are long, one might expect that it is difficult for recurrent networks to learn. We however find that with tuning, it is possible to train LSTM recurrent networks to fit the training set. For example, if we set the size of hidden state to be 512 units and truncate the backprop to be 400, an LSTM can do fairly well. With random embedding dimension dropout [38] and random word dropout (not published previously), we are able to reach performance of around 86.5% accuracy in the test set, which is approximately 5% worse than most baselines.

Fundamentally, the main problem with this approach is that it is unstable: if we were to increase the number of hidden units or to increase the number of backprop steps, the training breaks down very quickly: the objective function explodes even with careful tuning of the gradient clipping. This is because LSTMs are sensitive to the hyperparameters for long documents. In contrast, we find that the SA-LSTM works better and is more stable. If we use the sequence autoencoders, changing the size of the hidden state or the number of backprop steps hardly affects the training of LSTMs. This is important because the models become more practical to train.

Using sequence autoencoders, we overcome the optimization instability in LSTMs in such a way that it is fast and easy to achieve perfect classification on the training set. To avoid overfitting, we again use input dimension dropout, with the dropout rate chosen on a validation set. We find that dropping out 80% of the input embedding dimensions works well for this dataset. The results of our experiments are shown in Table 2 together with previous baselines. We also add an additional baseline where we initialize a LSTM with word2vec embeddings on the training set.

Table 2: Performance of models on the IMDB sentiment classification task.

| Model | Test error rate |
|---|---|
| LSTM with tuning and dropout | 13.50% |
| LSTM initialized with word2vec embeddings | 10.00% |
| LM-LSTM (see Section 2) | 7.64% |
| SA-LSTM (see Figure 1) | 7.24% |
| SA-LSTM with linear gain (see Section 3) | 9.17% |
| SA-LSTM with joint training (see Section 3) | 14.70% |
| Full+Unlabeled+BoW [22] | 11.11% |
| WRRBM + BoW (bnc) [22] | 10.77% |
| NBSVM-bi (Naïve Bayes SVM with bigrams) [36] | 8.78% |
| seq2-bow$n$-CNN (ConvNet with dynamic pooling) [12] | 7.67% |
| Paragraph Vectors [19] | 7.42% |

The results confirm that SA-LSTM with input embedding dropout can be as good as previous best results on this dataset. In contrast, LSTMs without sequence autoencoders have trouble in optimizing the objective because of long range dependencies in the documents.

Using language modeling (LM-LSTM) as an initialization works well, achieving 8.98%, but less well compared to the SA-LSTM. This is perhaps because language modeling is a short-term objective, so that the hidden state only captures the ability to predict the next few words.

In the above table, we use 1,024 units for memory cells, 512 units for the input embedding layer in the LM-LSTM and SA-LSTM. We also use a hidden layer 30 units with dropout of 50% between the last hidden state and the classifier. We continue to use these settings in the following experiments.

In Table 3, we present some examples from the IMDB dataset that are correctly classified by SA-LSTM but not by a bigram NBSVM model. These examples often have long-term dependencies or have sarcasm that is difficult to detect by solely looking at short phrases.

## 4.2 Sentiment analysis experiments with Rotten Tomatoes and the positive effects of additional unlabeled data

The success on the IMDB dataset convinces us to test our methods on another sentiment analysis task to see if similar gains can be obtained. The benchmark of focus in this experiment is the Rotten Tomatoes dataset [26].[2] The dataset has 10,662 documents, which are randomly split into 80% for training, 10% for validation and 10% for test. The average length of each document is 22 words and the maximum length is 52 words. Thus compared to IMDB, this dataset is smaller both in terms of the number of documents and the number of words per document.

Table 3: IMDB sentiment classification examples that are correctly classified by SA-LSTM and incorrectly by NBSVM-bi.

| Text | Sentiment |
|------|-----------|
| Looking for a REAL super bad movie? If you wanna have great fun, don't hesitate and check this one! Ferrigno is incredibly bad but is also the best of this mediocrity. | Negative |
| A professional production with quality actors that simply never touched the heart or the funny bone no matter how hard it tried. The quality cast, stark setting and excellent cinemetography made you hope for Fargo or High Plains Drifter but sorry, the soup had no seasoning...or meat for that matter. A 3 (of 10) for effort. | Negative |
| The screen-play is very bad, but there are some action sequences that i really liked. I think the image is good, better than other romanian movies. I liked also how the actors did their jobs. | Negative |

Our first observation is that it is easier to train LSTMs on this dataset than on the IMDB dataset and the gaps between LSTMs, LM-LSTMs and SA-LSTMs are smaller than before. This is because movie reviews in Rotten Tomatoes are sentences whereas reviews in IMDB are paragraphs.

As this dataset is small, our methods tend to severely overfit the training set. Combining SA-LSTMs with 95% input embedding and 50% word dropout improves generalization and allows the model to achieve 19.3% test set error.Tuning the SA-LSTM further on the validation set can improve the result to 19.3% error rate on the test set.

To better the performance, we add unlabeled data from the IMDB dataset in the previous experiment and Amazon movie reviews [23] to the autoencoder training stage.[3] We also run a control experiment where we use the pretrained word vectors trained by word2vec from Google News.

Table 4: Performance of models on the Rotten Tomatoes sentiment classification task.

| Model | Test error rate |
|-------|-----------------|
| LSTM with tuning and dropout | 20.3% |
| LM-LSTM | 21.9% |
| LSTM with linear gain | 22.2% |
| SA-LSTM | 19.3% |
| LSTM with word vectors from word2vec Google News | 20.5% |
| SA-LSTM with unlabeled data from IMDB | 18.6% |
| SA-LSTM with unlabeled data from Amazon reviews | 16.7% |
| MV-RNN [29] | 21.0% |
| NBSVM-bi [36] | 20.6% |
| CNN-rand [13] | 23.5% |
| CNN-non-static (ConvNet with word vectors from word2vec Google News) [13] | 18.5% |

The results for this set of experiments are shown in Table 4. Our observation is that if we use the word vectors from word2vec, there is only a small gain of 0.5%. This is perhaps because the recurrent weights play an important role in our model and are not initialized properly in this experiment. However, if we use IMDB to pretrain the sequence autoencoders, the error decreases from 20.5% to 18.6%, nearly a 2% gain in accuracy; if we use Amazon reviews, a larger unlabeled dataset (7.9 million movie reviews), to pretrain the sequence autoencoders, the error goes down to 16.7% which is another 2% gain in accuracy.

This brings us to the question of how well this method of using unlabeled data fares compared to adding more labeled data. As argued by Socher et al. [30], a reason of why the methods are not perfect yet is the lack of labeled training data, they proposed to use more labeled data by labeling an addition of 215,154 phrases created by the Stanford Parser. The use of more labeled data allowed their method to achieve around 15% error in the test set, an improvement of approximately 5% over older methods with less labeled data.

We compare our method to their reported results [30] on sentence-level classification. As our method does not have access to valuable labeled data, one might expect that our method is severely disadvantaged and should not perform on the same level. However, with unlabeled data and sequence autoencoders, we are able to obtain 16.7%, ranking second amongst many other methods that have access to a much larger corpus of labeled data. The fact that unlabeled data can compensate for the lack of labeled data is very significant as unlabeled data are much cheaper than labeled data. The results are shown in Table 5.

Table 5: More unlabeled data vs. more labeled data. Performance of SA-LSTM with additional unlabeled data and previous models with additional labeled data on the Rotten Tomatoes task.

| Model | Test error rate |
|---|---|
| LSTM initialized with word2vec embeddings trained on Amazon reviews | 21.7% |
| SA-LSTM with unlabeled data from Amazon reviews | 16.7% |
| NB [30] | 18.2% |
| SVM [30] | 20.6% |
| BiNB [30] | 16.9% |
| VecAvg [30] | 19.9% |
| RNN [30] | 17.6% |
| MV-RNN [30] | 17.1% |
| RNTN [30] | 14.6% |

### 4.3 Text classification experiments with 20 newsgroups

The experiments so far have been done on datasets where the number of tokens in a document is relatively small, a few hundred words. Our question becomes whether it is possible to use SA-LSTMs for tasks that have a substantial number of words, such as web articles or emails and where the content consists of many different topics.

For that purpose, we carry out the next experiments on the 20 newsgroups dataset [17].[4] There are 11,293 documents in the training set and 7,528 in the test set. We use 15% of the training documents as a validation set. Each document is an email with an average length of 267 words and a maximum length of 11,925 words. Attachments, PGP keys, duplicates and empty messages are removed. As the newsgroup documents are long, it was previously considered improbable for recurrent networks to learn anything from the dataset. The best methods are often simple bag-of-words.

We repeat the same experiments with LSTMs and SA-LSTMs on this dataset. Similar to observations made in previous experiments, SA-LSTMs are generally more stable to train than LSTMs. To improve generalization of the models, we again use input embedding dropout and word dropout chosen on the validation set. With 70% input embedding dropout and 75% word dropout, SA-LSTM achieves 15.6% test set error which is much better than previous classifiers in this dataset. Results are shown in Table 6.

### 4.4 Character-level document classification experiments with DBpedia

In this set of experiments, we turn our attention to another challenging task of categorizing Wikipedia pages by reading character-by-character inputs. The dataset of attention is the DBpedia dataset [20], which was also used to benchmark convolutional neural nets in Zhang and LeCun [39].

Table 6: Performance of models on the 20 newsgroups classification task.

| Model | Test error rate |
|---|---|
| LSTM | 18.0% |
| LM-LSTM | 15.3% |
| LSTM with linear gain | 71.6% |
| SA-LSTM | 15.6% |
| Hybrid Class RBM [18] | 23.8% |
| RBM-MLP [5] | 20.5% |
| SVM + Bag-of-words [3] | 17.1% |
| Naïve Bayes [3] | 19.0% |

Note that unlike other datasets in Zhang and LeCun [39], DBpedia has no duplication or tainting issues so we assume that their experimental results are valid on this dataset. DBpedia is a crowd-sourced effort to extract information from Wikipedia and categorize it into an ontology.

For this experiment, we follow the same procedure suggested in Zhang and LeCun [39]. The task is to classify DBpedia abstracts into one of 14 categories after reading the character-by-character input. The dataset is split into 560,000 training examples and 70,000 test examples. A DBpedia document has an average of 300 characters while the maximum length of all documents is 13,467 characters. As this dataset is large, overfitting is not an issue and thus we do not perform any dropout on the input or recurrent layers. For this dataset, we use a two-layered LSTM, each layer has 512 hidden units and and the input embedding has 128 units.

Table 7: Performance of models on the DBpedia character level classification task.

| Model | Test error rate |
|---|---|
| LSTM | 13.64% |
| LM-LSTM | 1.50% |
| LSTM with linear gain | 1.32% |
| SA-LSTM | 2.34% |
| SA-LSTM with linear gain | 1.23% |
| SA-LSTM with 3 layers and linear gain | 1.19% |
| SA-LSTM (word-level) | 1.40% |
| Bag-of-words | 3.57% |
| Small ConvNet | 1.98% |
| Large ConvNet | 1.73% |

In this dataset, we find that the linear label gain as described in Section 3 is an effective mechanism to inject gradients to earlier steps in LSTMs. This linear gain method works well and achieves 1.32% test set error, which is better than SA-LSTM. Combining SA-LSTM and the linear gain method achieves 1.19% test set error, a significant improvement from the results of convolutional networks as shown in Table 7.

### 4.5 Object classification experiments with CIFAR-10

In these experiments, we attempt to see if our pre-training methods extend to non-textual data. To do this, we train a LSTM to read the CIFAR-10 image dataset row-by-row (where the input at each timestep is an entire row of pixels) and output the class of the image at the end. We use the same method as in [16] to perform data augmentation. We also trained a LSTM to do next row prediction given the current row (we denote this as LM-LSTM) and a LSTM to predict the image by rows after reading all its rows (SA-LSTM). We then fine-tune these on the classification task. We present the results in Table 8. While we do not achieve the results attained by state of the art convolutional networks, our 2-layer pretrained LM-LSTM is able to exceed the results of the

baseline convolutional DBN model [15] despite not using any convolutions and outperforms the non pre-trained LSTM.

Table 8: Performance of models on the CIFAR-10 object classification task.

| Model | Test error rate |
|---|---|
| 1-layer LSTM | 25.0% |
| 1-layer LM-LSTM | 23.1% |
| 1-layer SA-LSTM | 25.1% |
| 2-layer LSTM | 26.0% |
| 2-layer LM-LSTM | 18.7% |
| 2-layer SA-LSTM | 26.0% |
| Convolution DBNs [15] | 21.1% |

## 5   Discussion

In this paper, we found that it is possible to use LSTM recurrent networks for NLP tasks such as document classification. We also find that a language model or a sequence autoencoder can help stabilize the learning in recurrent networks. On five benchmarks that we tried, LSTMs can become a general classifier that reaches or surpasses the performance levels of all previous baselines.

**Acknowledgements:**   We thank Oriol Vinyals, Ilya Sutskever, Greg Corrado, Vijay Vasudevan, Manjunath Kudlur, Rajat Monga, Matthieu Devin, and the Google Brain team for their help.

## Footnotes

[1] http://ai.Stanford.edu/amaas/data/sentiment/index.html

[2]http://www.cs.cornell.edu/people/pabo/movie-review-data/

[3]The dataset is available at `http://snap.stanford.edu/data/web-Amazon.html`, which has 34 million general product reviews, but we only use 7.9 million movie reviews in our experiments.

[4]http://qwone.com/~jason/20Newsgroups/

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
