[Reviews · NeurIPS 2015]

Submitted by Assigned_Reviewer_1

[ I have read the authors' response ]

This is a well-written paper with a clear message: pre-training LSTMs to auto-encode improves performance of the LSTM on a supervised learning task.

This method is compared to several other methods for pre-training, and evaluated on 5 datasets spanning 3 domains.

The interesting result is that LSTMs with random initialization generally do worse than state-of-the-art methods, whereas pre-training with self-encoding enables LSTMs to do better than diverse past methods on a variety of tasks.

The absolute improvements are respectable, but what is interesting is that a single method yields state-of-the-art performance on a collection of tasks.

I wasn't clear how much the network setup differed between the SA-LSTM experiments in different domains.

Were the number of hidden units, network topology, and any free parameters identical?

In section 4.3, what is being predicted?

Overall, because the message is clear, the method is easy to understand and implement, and the results are consistent across domains, I think others will adopt this method.

I believe this papers is destined to have impact.
Summary: This clearly written, experimentally sound paper shows that pre-training LSTMs to auto-encode improves performance of the LSTM on a variety of supervised learning tasks.

Submitted by Assigned_Reviewer_2

This is a mostly empirical paper where the authors apply a new style of pretraining to LSTMs and show major improvements over randomly initialized LSTM on sentiment analysis, text classification, and character-level document classification. The main pretraining approach is to simply train the LSTM as auto-encoder to be able to echo a sequence in order.

Generally this paper shows very striking results with gains over temporal convolutional networks and document vector methods like paragraph vectors.

General comments:

- "Unsupervised Sequence Learning" is a confusing title for this

paper. While the pretraining method does use an unsupervised

sub-task, there are no results reported on any unsupervised tasks

or even analysis of this initial subtask.

- The LSTM autoencoder itself (not for pretraining) does not seem

novel. The seq2seq work for instance uses "unsupervised"

reconstruction as an example (particularly in their talks).

This

should be made clear in the paper.

- I was not able to follow the explanation in paragraph l.97

- l. 182 "Random word dropout" seems important. Could you describe?

- It was not clear exactly how pretraining was done. Unless I am

mistaked, it seems like the autoencoder is trained on in-domain

documents possibly from the same dataset? Is this important? What

happens if you pretrain on wikipedia or external data. For section

4.3 how is SA pretrained?

- l.243. It seems unfair that CNN-non-static is not given the Amazon

reviews as well. Should be possible to run word2vec on amazon

instead of google news for this comparison.

- One aspect that would improve this paper is some direct evidence of

the claim that long-range dependencies are really being captured. It

seems possible that the LSTM is just somehow better than CNN at

picking up local phrases. Even anecdotal evidence of this process

would be interesting.

## Small

- l103: typo "the weights" verb agreement - l114: typo verb agreement. - l188: same
Summary: This paper shows striking empirical evidence that LSTMs with simple pretraining can outperform CNNs on a variety of text classification tasks.

Submitted by Assigned_Reviewer_3

Deep long-short term models (LSTMs) can suffer from initialization, gradient problems, and lack of labeled data. The extremely elegant solution to this problem is an autoencoder which trains the models to output the exact same input with partial information. This allows for stronger initialization parameters for labeled tasks as well as gradient short cuts.

Parameter initialization is a large problem with LSTMs. The autoencoder solution is natural and makes sense. The author should address in more detail why LSTMs with linear gain perform so poorly on the 20 newsgroup classification task. In the Rotten Tomatoes task, further clarification as to why SA-LSTM does not have access to the same labeled data would be helpful. Further mathematical explaination of the gradient shortcut would also be helpful
Summary: The authors' intuition about sequence autoencoders is extremely simple and elegant. The results are interesting and impactful.

Submitted by Assigned_Reviewer_4

This paper addresses the issue of how to best pretrain an LSTM when used as first module in an classification task.

Two methods are proposed: training an LSTM language model or training a sequence to sequence auto-encoder. In both cases, the LSTM can be trained on the data labeled for the classification task, but also on additional unlabeled data.

The idea is thoroughly evaluated on five different tasks, achieving state-of-the-art results in each task. This is mainly experimental research. The paper is clearly written and easy to read.

Overall, I have one theoretical concern. When pre-training some modules of an classifier with unsupervised data, in particular in the case of an auto-encoder, this module aims in preserving ALL information of the input.

However, parts of this information may be irrelevant for the final classification task itself (e.g. all the stop and filler words are memorized although probably completely irrelevant for a sentiment classification task). This need for preserving all information may push to the use of LSTMs with large capacity, which need to be carefully regularized. In some of the experiments, rather high drop-out values of up to 80% are used.

Finally, I think that the results on the MT task are too preliminary to make strong conclusions. Perplexity is not the usual metric to measure the quality of an MT system. Anyway, the observed improvements are less than 3% relative.
Summary: This papers considers 5 different sequence classification tasks. Important gains are obtained by pretraining the LSTM as an language model or seq2seq auto-encoder. Both approaches allow the use on unlabelled data.

Author Feedback
Author rebuttal: We thank the reviewers for the constructive comments and feedback. In the following, we will address some of the main concerns by the reviewers.

Assigned_Reviewer_1:

- The reviewer is concerned with whether the models are different. We can confirm that all word-level models have the same architecture (same number hidden units, size of embeddings, learning rates, weight decay). The translation model has a different architecture because the dataset is much bigger.

- Regarding the question about section 4.3, we can confirm that the model is predicting which of the 20 newsgroups an email comes from.

Assigned_Reviewer_2:

- The reviewer is concerned with the fact that linear label gain performs poorly on the newsgroup task. We observe that the newsgroup classification task has much higher variance in document length compared to the other datasets. We suspect that linear gain may cause too much weight to be put on words earlier in the document when encountering long documents.

- The reviewer asks why we did not use more labeled data. Our explanation is that additional labels in (Socher et al, 2013) are short phrase-level labels that were collected intentionally for Recursive models. We seek to show that this kind of additional annotation is unnecessary.

Assigned_Reviewer_3:

- The reviewer has a theoretical concern about the method. We agree that one can always contrive an artificial case where pretraining should not help. We also agree that our contribution has been to show empirical evidence that pretraining for text classification is a good idea. The network architectures in the experiments are the same to avoid unfair tuning.

- In order to understand the relationship between auto-encoding, model sizes, and text classification, we ran an additional experiment with a smaller network of 256 hidden units in the LSTM (80% reduction in terms of hidden layer size) and lowered the dropout rate to 50%. With this, the network still managed to achieve 90.5% accuracy (vs 88.2% with random initialization) on IMDB (the input dictionary is 180k). This means that when the network is unable to preserve ALL information, the network chooses to to focus efforts on topical words which have signatures of the document. These topical words happen to pertain to most supervised labels we care about (such as topic or sentiment of documents). The fact that the pretraining keeps key words of the document and the classification task requires this information is the main reason why the pretraining works. If the classification task were about counting the number of words in the document, there is no reason for pretraining at all.

- In addition to that, the fact that language modeling works for texts when the number of hidden units is low (see Mikolov et al, 2013) proves that the manifold of language indeed has low intrinsic dimensionality. Since the hidden states are often small, the autoencoders or language models learn regularities of documents (and remove noisy dimensions). The success of these models as a pretraining method perhaps demonstrates that keeping the regularity structures of language can help subsequent classification tasks.

- We believe that the MT results are strong but we are happy to remove the MT results. We provided results on perplexity because in Neural Machine Translation, perplexity is well correlated with human evaluation and BLEU scores (see Luong et al, ACL 2015).

Assigned_Reviewer_4:
- The reviewer is concerned with the unfairness of the experiments on word2vec initialization. We agree, and will provide more results in the final revision of the paper.

- We will think more about the title, and give better explanations to the mentioned sections.

- The pretraining was done on the same dataset of the classification task for all results except table 4 and the middle section of table 3. For those tables, we show that using external unlabelled data can improve performance on the supervised task.

- In paragraph l97, if in figure 1, after the input < eos > token, the subsequent inputs are padding tokens. In this case, any gradients flowing to the embedding for 'W' would have to pass through at least 5 timesteps instead of 1. By random word dropout, when we process a training example, we randomly delete a subset of the words.

- We plan to add examples of long range dependencies that the LSTM is able to catch that the other models cannot.

Assigned_Reviewer_6:

- We didn't find the paper particularly relevant to our work but we are happy to cite it.

- The reviewer is concerned about other ways of initializing or obtaining embeddings. We plan to do experiments with word2vec embeddings trained on Amazon to augment table 3. We did perform analysis of initializing the LSTM with either the SA-LSTM embeddings, the SA-LSTM weights or just the variance of the weights. None of these methods performed as well as initializing from the entire SA-LSTM.